# Comparison of Patient Survival According to Erythropoiesis-Stimulating Agent Type of Treatment in Maintenance Hemodialysis Patients

**DOI:** 10.3390/jcm12020625

**Published:** 2023-01-12

**Authors:** Seok Hui Kang, Bo Yeon Kim, Eun Jung Son, Gui Ok Kim, Jun Young Do

**Affiliations:** 1Division of Nephrology, Department of Internal Medicine, College of Medicine, Yeungnam University, Daegu 42415, Republic of Korea; 2Healthcare Review and Assessment Committee, Health Insurance Review and Assessment Service, Wonju 26465, Republic of Korea; 3Quality Assessment Department, Health Insurance Review and Assessment Service, Wonju 26465, Republic of Korea

**Keywords:** erythropoiesis-stimulating agent, hemodialysis, mortality, outcome, survival

## Abstract

This study aimed to evaluate the difference in patient survival according to the type of erythropoiesis-stimulating agent (ESA) treatment used in the Korean hemodialysis (HD) population. This retrospective study analyzed the laboratory data from a national HD quality assessment program and the claims of Korea. Included participants were divided into three groups according to the type of ESA used during the 6 months of each assessment period as follows: the EP group (*n* = 38,043, epoetin-α or epoetin-β), the DP group (*n* = 10,054, darbepoetin-α), and the MR group (2253, continuous erythropoietin receptor activator). The ESA doses in the EP, DP, and MR groups were 6451 ± 3586, 5959 ± 3857, and 3877 ± 2275 unit/week, respectively. The erythropoiesis resistance indexes (ERIs) in the three groups were 10.7 ± 6.7, 9.9 ± 7.6, and 6.3 ± 4.1 IU/kg/g/dL, respectively. Kaplan–Meier curves revealed similar rates of patient survival among the three groups (*p* = 0.530). A multivariate Cox regression analysis showed that the hazard ratios in the DP group and MR group were 1.00 (*p* = 0.853) and 0.87 (*p* < 0.001), respectively, compared to that of the EP group. The hazard ratio in the MR group was 0.87 (*p* = 0.001) compared to that of the DP group. Our study shows that the MR group had comparable or better patient survival than the EP and DP groups in the multivariate analysis. However, the ESA doses and ERI were considerably different among the three groups. It was difficult to determine whether the better patient survival in the MR group originated from the ESA type, ESA dose, ERI, or other hidden factors.

## 1. Introduction

Chronic kidney disease (CKD) is an increasing health problem, with a prevalence of approximately 11.1% in Korea and 14.9% in the USA [1,2]. CKD is divided into five stages according to progression. The most advanced stage of CKD requires renal replacement therapies, including hemodialysis (HD), peritoneal dialysis, or kidney transplantation. HD is the most commonly used among the three modalities of renal replacement therapy, and the HD rates among patients who have undergone renal replacement therapy are 81% in Korea and 67% in USA [2,3]. It has been found that most patients who undergo HD are prone to anemia, developed by a uremic toxin, abnormal iron status, and decreased erythropoietin production [4]. An erythropoiesis-stimulating agent (ESA) is the main option for anemia treatment in HD patients. ESAs can be divided into three types according to half-life or dosing interval (short-acting treatment three times per week, intermediate-acting treatment once per week, and long-acting treatment once per 2 weeks to 1 month).

ESA binds the ESA receptor, and the signaling pathway for erythropoiesis after binding is not different among the three ESAs. However, some concerns about patient outcomes according to the three ESAs exist. Some studies have failed to identify the association between the types of ESAs and clinical outcomes, but other studies have reported favorable or unfavorable results with a particular type of ESA and involved inconsistent results with no robust evidence [5,6,7,8]. Further, results may differ according to ethnicity or region, mostly due to the national policy, the cost of the selection of the drug, the efficacy of the ESA types, or differences in the ESA dose as per body size. Therefore, nation- and ethnic-specific results would be useful in determining the associations between the types of ESAs and clinical outcomes in HD patients. However, in Korea, only a few studies are available in this regard. This study aimed to evaluate the difference in patient survival according to the type of ESA in the Korean HD population.

## 2. Materials and Methods

### 2.1. Data Source

This retrospective study analyzed the laboratory data from a national HD quality assessment program and the claims data from the Health Insurance Review & Assessment (HIRA) of the Republic of Korea [9,10]. The fourth, fifth, and sixth HD quality assessment programs were performed in July 2013–December 2013, July 2015–December 2015, and March 2018–August 2018, respectively. The programs included HD patients on maintenance for ≥3 months, those undergoing HD at least twice a week (≥8 times monthly), and those aged ≥18 years.

In addition, we analyzed the claims data of all HD patients who underwent an HD quality assessment by HIRA. The Korean national healthcare system and the Medical Aid program cover almost the entire South Korean population. The HIRA, as a government-affiliated organization, has nearly all medical information of patients, from diagnoses and past medical records to procedural data. The numbers of patients included in the fourth, fifth, and sixth HD quality assessment programs were 21,846, 35,538, and 31,294, respectively. We excluded repeated participants and those with an insufficient dataset (75 in the fourth, 13,795 in the fifth, and 18,570 in the sixth HD quality assessment programs) and those not using ESA for 6 months (*n* = 5888). Active bleeding correlates with high erythropoiesis resistance index (ERI) levels and patient survival. In this study, none of the patients received blood transfusions during the 6 months of the HD quality assessment. Therefore, this was considered to mean that none of the patients experienced active bleeding during the period. Finally, 50,350 patients were included in our study. The study was approved by the institutional review board of Yeungnam University Medical Center (approval no: YUMC 2022-01-010). All experiments were performed in accordance with the relevant guidelines and regulations. Informed consent was not obtained from the patients since the records and information of the participants were anonymized and de-identified before the analyses. The institutional review board of Yeungnam University Medical Center also waived the need for obtaining informed consent.

### 2.2. Study Population and Variables

The included participants were divided into three groups according to the type of ESA used during the 6 months of each HD quality assessment period as follows: EP, DP, and MR groups. Approximately 88% of our cohort used one ESA agent, and the remainder used two or more ESAs. The patients who used epoetin-alpha or epoetin-beta were included in the EP group (short-acting ESA). The patients who used darbepoetin-alpha were included in the DP group (intermediate-acting ESA). The patients who used a continuous erythropoietin receptor activator (CERA, long-acting ESA) were included in the MR group. The patients who used two or more ESAs were included in the ESA group according to the highest doses taken during the 6 months. The doses of the different ESAs were converted as the same unit (IU/week) using a conversion ratio from a previous study [11].

The HD quality assessment data were collected using a web-based data collection system. The data collected comprised age (years), sex, body mass index (kg/m^2^), the underlying etiology of end-stage renal disease, HD vintages (days), and the type of vascular access. The laboratory data from the assessment included hemoglobin (g/dL), Kt/V_urea_, serum albumin (g/dL), serum calcium (mg/dL), serum phosphorus (mg/dL), serum creatinine (mg/dL), pre-dialysis systolic blood pressure (SBP in mmHg), pre-dialysis diastolic blood pressure (DBP in mmHg), and ultrafiltration volume (L/session). These data were collected monthly and were averaged from monthly obtained values. Kt/V_urea_ was calculated using Daugirdas’ equation [12]. ESA dose (IU/week) was averaged from the total dose of ESA taken for 6 months. The ERI was calculated by using the following equation: ERI = ESA dose (IU/week)/body weight (kg)/hemoglobin level (g/dL) [13].

The presence of comorbidities was evaluated from 1 year before evaluating the HD quality assessment program and was defined using the codes used by Quan et al. [14,15]. Finally, the Charlson comorbidity index (CCI) score was calculated using scores from a previous study [14]. During the follow-up, clinical outcomes, except for death, were defined using electronic data. The codes were as follows: O7072, O7071, and O7061 for peritoneal dialysis, and R3280 for kidney transplantation.

### 2.3. Statistical Analyses

The data were analyzed using SAS Enterprise Guide version 7.1 (SAS Institute, Cary, NC, USA) or R version 3.5.1 (R Foundation for Statistical Computing, Vienna, Austria). Categorical variables are presented as numbers and percentages, while continuous variables are presented as mean ± standard deviation. The Pearson χ^2^ test or Fisher’s exact test was used to analyze categorical variables. For continuous variables, the means were compared using a one-way analysis of variance, followed by the Tukey post hoc test. The survival estimates were calculated using Kaplan–Meier curves and Cox regression analyses. *p*-values for the comparisons of the survival curves were determined using the log-rank test. Multivariate Cox regression analyses were adjusted for age, sex, CCI score, underlying the etiology of end-stage renal disease, body mass index, hemoglobin levels, serum creatinine, SBP, DBP, serum calcium, serum phosphorus, Kt/V_urea_, serum albumin, HD vintage, ultrafiltration volume, ERI, and ESA dose, and they were performed using the enter mode. For linear regression analyses, the dependent variable was ERI, whereas the independent variables were age; sex; CCI score; the underlying etiology of end-stage renal disease; body mass index; SBP; DBP; hemoglobin, serum creatinine, serum calcium, serum phosphorus and serum albumin levels; Kt/V_urea_; HD vintage; ultrafiltration volume; and ESA dose. *p* < 0.05 was considered statistically significant.

## 3. Results

### 3.1. Participant Clinical Characteristics

The numbers of patients in the EP, DP, and MR groups were 38,043, 10,054, and 2253, respectively. The patients’ baseline characteristics are shown in Table 1. 

The EP group had the highest proportion of men, arteriovenous fistula, HD vintage, ultrafiltration volume, serum albumin, serum calcium, serum creatinine, and DBP and the lowest age, Kt/V_urea_, and hemoglobin among the three groups. The DP group had the lowest follow-up duration and phosphorus. The MR group had the highest CCI score and body mass index. The ESA doses in the EP, DP, and MR groups were 6451 ± 3586, 5959 ± 3857, and 3877 ± 2275 unit/week, respectively (*p* < 0.001). The ERIs in the three groups were 10.7 ± 6.7, 9.9 ± 7.6, and 6.3 ± 4.1 IU/kg/g/dL, respectively (*p* < 0.001). Among the three groups, the ESA doses and ERIs were the lowest in the MR group.

### 3.2. Survival Analyses

The numbers of patient deaths, transfer to peritoneal dialysis, and kidney transplantation at the end-point of the follow-up were 12,804 (31.8%), 133 (0.3%), and 2762 (7.3%) in the EP group; 3028 (30.1%), 46 (0.5%), and 790 (7.9%) in the DP group; and 709 (31.5%), 8 (0.4%), and 152 (6.7%) in the MR group, respectively (*p* = 0.015). The Kaplan–Meier curve revealed no significant difference in patient survival among the three groups (Figure 1, *p* = 0.530).

The univariate Cox regression analysis showed that the hazard ratios in the DP and MR groups were 1.01 (95% confidence interval (CI), 0.98–1.05; *p* = 0.527) and 0.97 (95% CI, 0.91–1.04; *p* = 0.433), respectively, compared to that of the EP group (Table 2).

The hazard ratio in the MR group was 0.96 (95% CI, 0.89–1.04; *p* = 0.302) compared to that in the DP group. The multivariate Cox regression analysis showed that the hazard ratios in the DP group and MR group were 1.01 (95% CI, 0.97–1.05; *p* = 0.740) and 0.89 (95% CI, 0.83–0.97; *p* = 0.006), respectively, compared to that of the EP group. The hazard ratio in the MR group was 0.89 (95% CI, 0.82–0.97; *p* = 0.006) compared to that of the DP group.

### 3.3. Subgroup Analyses

The numbers of patients aged <65 years, patients aged ≥65 years, male patients, and female patients were 29,063, 21,287, 29,552, and 20,798, respectively. Figure 2 shows the Kaplan–Meier curves of the various subgroups of the three ESA groups. 

For patients aged <65 years, there was no significant difference in patient survival among the EP, DP, and MR groups in the univariate and multivariate analyses results (Appendix A). For those aged ≥65 years, the MR group had better patient survival than the EP and DP groups as per multivariate analyses (Appendix A). For men, the MR group had better patient survival than the DP group in the multivariate analysis (Appendix A). For women, the MR group had the best patient survival among the three groups in the multivariate analysis (Appendix A).

Additionally, we performed survival analyses according to ESA dose tertiles in the same types of ESAs (Figure 3).

The ESA doses in the low, middle, and high tertiles were 2916 ± 1350, 6122 ± 711, and 10,316 ± 2936 IU/week in the EP group; 2787 ± 994, 5187 ± 658, and 9905 ± 4122 IU/week in the DP group; and 1874 ± 649, 3158 ± 429, and 6241 ± 2277 IU/week in the MR group, respectively. The hemoglobin levels in the low, middle, and high tertiles were 10.8 ± 0.8, 10.6 ± 0.6, and 10.3 ± 0.7 g/dL in the EP group (*p* < 0.001); 11.0 ± 0.6, 10.7 ± 0.6, and 10.3 ± 0.8 g/dL in the DP group (*p* < 0.001); and 11.0 ± 0.6, 10.7 ± 0.7, and 10.3 ± 0.8 g/dL in the MR group (*p* < 0.001), respectively. The Kaplan–Meier curves showed that, in the EP and DP groups, patient survival decreased as the ESA dose tertile increased, and in the MR group, the high tertile had poorer patient survival than the low and middle tertiles. Cox-regression analyses showed that patient mortality increased as the tertiles increased in the EP group, but statistical significance was only obtained in the univariate analysis (Appendix A). These trends were the same as those in the DP group (Appendix A). In the MR group, the high tertile had the highest mortality among the three tertiles in the univariate analysis, but the high tertile in the MR group had a higher mortality rate than the middle tertile alone in multivariate analysis (Appendix A).

### 3.4. Factors Associated with ERI

Linear regression analyses were performed to identify the factor associated with ERI (Appendix A). The multivariate analysis showed that age, sex, CCI score, the underlying etiology of end-stage renal disease, SBP, Kt/V_urea_, HD vintage, and ESA dose were positively associated with ERI. Body mass index; hemoglobin, serum creatinine, calcium, phosphorus, and serum albumin levels; and ultrafiltration volume were inversely associated with ERI.

## 4. Discussion

Our study included 50,350 HD patients who underwent an HD quality assessment program. There were significant differences in the baseline characteristics among the three groups, but the differences in most variables were small. The ESA dose and ERI were significantly lower in the MR group than in the EP and DP groups. There was no significant difference in patient survival among the three groups in the univariate analysis, but the MR group had the highest patient survival rate among the three groups in the multivariate analysis results. Subgroup analyses by age and sex showed that the univariate analysis did not show a robust difference in patient survival among the three groups, but the multivariate analysis showed favorable survival in the MR group, especially in patients aged ≥ 65 years and female patients. We also analyzed the dose effect on patient survival within the same ESA group. The univariate analysis showed a positive association between the ESA dose tertiles and patient mortality, but statistical significance in the multivariate analysis was only obtained for the high and middle tertiles in the MR group.

In our study, there were large differences in the ESA dose and ERI according to the main ESA types, which were associated with ERI and the ESA dose for optimal hemoglobin level. The half-life of ESA is inversely associated with an affinity for the erythropoietin receptor, but a low affinity for the receptor is associated with low receptor occupancy [16]. Therefore, long-acting ESAs with a low affinity for receptors have been found to be poorly effective in patients with ESA resistance, and they have been associated with a slow increase in hemoglobin levels. These findings reveal that patients requiring a high ESA dose will benefit more from the administration of short- or intermediate-acting ESAs than long-acting ESAs, and the real-world data from our study also show similar trends.

In this study, the survival benefit in the MR group was more prominent in the multivariate analysis results than in the univariate analysis, but this may be associated with confounding effects, such as age and comorbidity. The MR group was older and had a greater CCI score than the EP and DP groups. Unfavorable factors in the MR group would lead to attenuating the survival benefit in the MR group, but adjustment for these factors independently showed better survival rates in the MR group than in the EP and DP groups. In addition, we suggest that the survival benefit in the MR group may not be caused by the class effects of the different ESAs. Our results should be interpreted by considering the differences in the baseline characteristics, ESA dose, ERI, and non-significance in the univariate analysis. The better survival in the MR group might be associated with the indirect effects of the prescription patterns of ESAs based on the requirement of high ESA doses or ERI rather than the class effects of the different ESAs.

In addition, our study evaluated patient survival according to the ESA dose tertiles in the same ESA group. The univariate analysis revealed that the patient mortality increased as the ESA dose tertile increased in all groups. However, the multivariate analysis showed a modest difference in the high tertile alone in the MR group. Previous studies have reported a positive association between high doses of ESAs and adverse outcomes, such as all-cause mortality and cardiovascular disease [5,6,7]. However, it is unclear whether the association between high doses of ESAs and adverse outcomes is caused by the higher ESA doses required for maintaining higher hemoglobin levels or the direct effect of the ESA dose. In our data, the mean hemoglobin level decreased and ERI increased as the ESA dose tertile in the same ESA group increased. These results reveal that ERI strongly influences the ESA dose, and direct effects on patient survival caused by various factors associated with ERI would lead to the attenuation of the direct effect of the ERA dose. 

Inconsistent results regarding the association between the types of ESAs and survival in patients with CKD have been obtained. A meta-analysis included randomized studies examining patients at any CKD stage and evaluated the effect on the outcomes of CERA compared to that on the outcomes of other ESAs [5]. The study did not show statistical significance in all-cause mortality between CERA and other ESAs. Sakaguchi et al. evaluated a nationwide cohort (*n* = 194,698) and found that patient survival was poor in those taking long-acting ESAs (darbepoetin-alfa and CERA) compared to in those taking short-acting ESAs (epoetin-alfa/beta/kappa) [6]. However, a randomized trial after Sakaguchi’s study did not show significant differences in patient survival between those taking long-acting ESAs and those taking short-acting ESAs [7]. Locatelli et al. performed a non-inferiority trial between CERA and other ESAs [7]. The study enrolled patients at any stage of CKD, but 84% of the total patients underwent dialysis. There was no significant difference in all-cause mortality between the two groups. Minutolo et al. performed a retrospective observational study that enrolled non-dialysis patients with CKD and compared short-acting and long-acting ESAs [8]. However, the study did not find differences in all-cause mortality between the two groups. The result discrepancy between Sakaguchi’s study and Locatelli’s and Minutolo’s studies has been interpreted to be due to the differences in ESA doses in patients treated with long-acting ESAs [7,8]. In our study, the mean ESA dose in the MR group was 3877 IU/week (64 μg/month as CERA). The mean or median ESA dose was 110.6 μg/month in Sakaguchi’s study, 75.2–112 μg/month in Locatelli’s study, and 75 μg/month in Minutolo’s study. The ESA dose in the MR group in our study was lower than those in these previous studies, which may be associated with the better patient survival in the MR group in our study.

Micronutrients influence iron status and HD patients’ responses to ESAs and survival. The micronutrient levels of HD patients are lower than those of the general population. This decrease correlates with decreasing gastrointestinal absorption and losses during HD [17]. Micronutrients, such as zinc, selenium, and magnesium, are well-known cofactors for enzymatic systems, and they play key roles in maintaining antioxidant activity. Although the pathophysiology of the roles of micronutrients in HD patients is poorly understood, epidemiological studies have shown positive associations between micronutrients and HD patient outcomes. A cross-sectional study evaluated the association between the response to ESAs and selenium levels in HD patients in Japan [18]. Their study showed an inverse association in response to ESAs based on selenium levels, and this association was independent of iron status. A retrospective observational study, which included 85 dialysis patients, showed a positive association between decreased selenium levels and mortality in dialysis patients [19].

Kobayashi et al. evaluated the importance of zinc in HD patients’ responses to ESAs [20]. HD patients were randomly divided into two groups: (1) patients without zinc replacement and (2) patients with zinc replacement. Significant improvements in the response to ESAs and iron status were observed in patients with zinc replacement compared to patients without zinc replacement. A multicenter prospective study evaluated the association between clinical outcomes and dietary zinc intake in HD patients [21]. The results showed that dietary zinc intake was correlated with nutritional status, body composition, and mortality in HD patients.

Other studies that have evaluated the association between magnesium and mortality in dialysis patients and a meta-analysis using 21 studies revealed a positive association between the level of magnesium and all-cause and cardiovascular mortality in dialysis patients [22]. Yu et al. also showed an association between magnesium levels and HD patients’ responses to ESAs [23]. Previous studies have also suggested that improvements in insulin sensitivity, serum albumin levels, endothelial function, and the energy metabolism of erythrocytes may contribute to improvements in the responsiveness of ESAs via magnesium [24,25,26]. Our study did not include data on micronutrients. Therefore, an analysis of micronutrients is beyond the scope of our study. Even so, we identified that the response to the ESAs and mortality had significant differences according to the ESA type. Variations in micronutrient levels may account for the observed differences in the response to the ESAs and mortality in HD patients. Further studies on micronutrients are required to elucidate the cause-and-effect relationships for the observed differences in the response to the ESAs and mortality according to the ESA types.

Factors associated with ERI are essential issues in the management of these patients. Our results revealed that ERI positively correlates with age, female sex, HD vintage, and comorbidities. In addition, we observed that ERI is inversely related to direct or indirect nutritional indicators, such as serum albumin levels, phosphorus, body mass index, serum creatinine levels, and ultrafiltration volume. A recent guideline stated that factors such as iron deficiency, vitamin deficiency, inflammation, bleeding, hyperparathyroidism, and malnutrition are correlated with the hypo-responsiveness of ESA (high ERI) [3]. González-Ortiz et al. showed a positive association between protein–energy wasting and ERI in HD patients [27]. Other studies have shown that the geriatric nutritional risk index, body composition, and sex are associated with ERI in HD patients [28,29,30]. Comorbidities have variable effects on the responsiveness of ESA, which are mediated through the inflammatory response and leads to increases in ERI [30]. Our results indicate a positive association between direct and indirect nutritional indicators and ERI. An association between comorbidities and ERI is congruent with findings from previous studies. Although the factors associated with ERI were beyond the scope of our study, our results may aid in identifying an independent factor for ERI.

Our study had limitations (Table 3).

First, this was a retrospective observational study that analyzed a dataset covering the HD quality assessment period and claims data without a medical chart review. The types and doses of the ESAs were evaluated using claims data. A discrepancy between the ESA prescriptions and the real doses may be present, and the route of injection was not evaluated. There are significant differences in the efficacy and half-life between subcutaneous and intravenous injections of ESAs. In addition, we did not exclude center or physician factors, such as ESA selection preferences. Second, our study did not evaluate the doses of iron supplements or iron status. In the fourth HD quality assessment, data on iron status was collected, but these data were not collected in the fifth or sixth HD quality assessment. Large proportions of patients did not have laboratory data on iron status, and we did not evaluate iron status. Iron status can influence ESA doses and hemoglobin levels, which can be important confounding factors. Further, various factors associated with ERI were not evaluated, and this is also an important confounding factor. Third, there were significant differences in sample sizes and baseline characteristics among the three groups, and this is associated with selection bias. Fourth, some patients used two or more ESAs within each group, but the proportions of these patients were relatively small (12%). Our data included CCI associated with comorbidity. However, CCI cannot completely represent the comorbidity status, and data on inflammatory status, iron status, history of bleeding, and safety issues related to ESA treatment were not obtained in our study. These data are important confounding factors for comparisons of the three groups through differences in baseline characteristics, ESA selection, and ERI values. Since our results were not adjusted for these variables, the risk of biased results cannot be excluded.

## 5. Conclusions

The present study found that the MR group had comparable or better patient survival than the EP and DP groups in the multivariate analysis. However, the ESA doses and ERIs were considerably different among the three groups. Considering the limitations of our study, it is difficult to determine whether the better patient survival in the MR group originated from the ESA type, ESA dose, ERI, or other hidden factors. Therefore, a prospective randomized cohort study that includes additional laboratory data should be conducted to overcome these limitations.

## Figures and Tables

**Figure 1 jcm-12-00625-f001:**
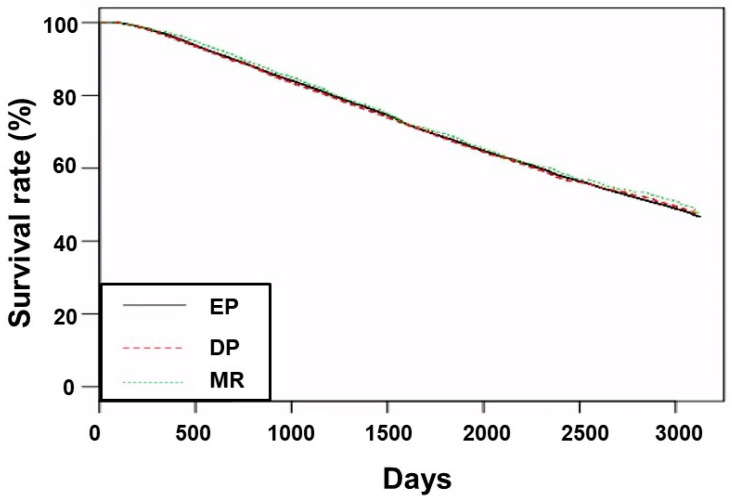
Kaplan–Meier curves of patient survival according to group. Abbreviations: EP, group treated with short-acting erythropoiesis-stimulating agents; DP, group treated with intermediate-acting erythropoiesis-stimulating agents; MR, group treated with long-acting erythropoiesis-stimulating agents.

**Figure 2 jcm-12-00625-f002:**
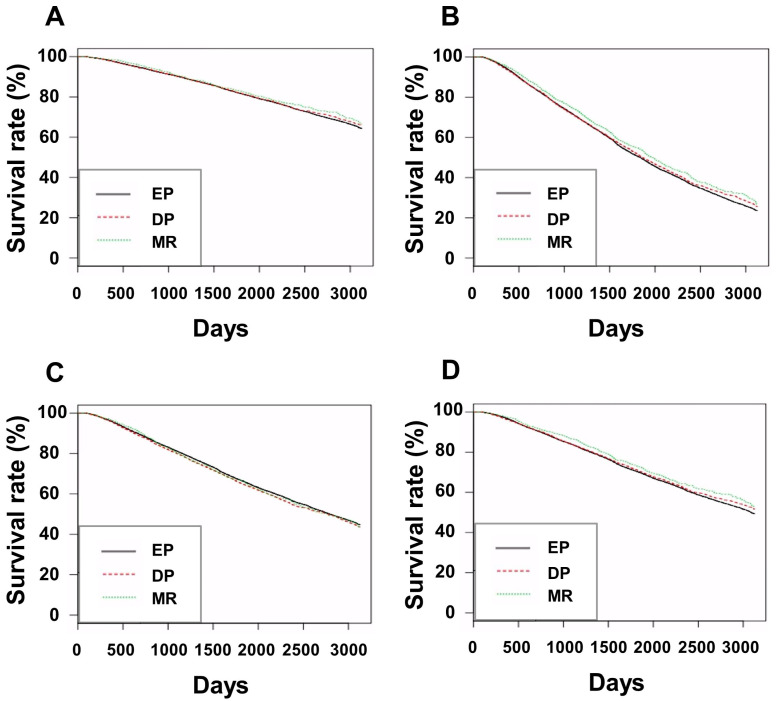
Kaplan–Meier curves of patient survival divided by patients aged <65 years, (**A**), patients aged ≥ 65 years, (**B**), male patients (**C**), and female patients (**D**). For patients aged < 65 years, *p*-values were 0.500 for EP vs. DP group, 0.500 for EP vs. MR group, and 0.500 for DP vs. MR group. For patients aged ≥ 65 years, *p*-values were 0.244 for EP vs. DP group, 0.036 for EP vs. MR group, and 0.118 for DP vs. MR group. For male patients, *p*-values were 0.087 for EP vs. DP group, 0.615 for EP vs. MR group, and 0.615 for DP vs. MR group. For female patients, *p*-values were 0.230 for EP vs. DP group, 0.160 for EP vs. MR group, and 0.230 for DP vs. MR group. Abbreviations: EP, group treated with short-acting erythropoiesis-stimulating agents; DP, group treated with intermediate-acting erythropoiesis-stimulating agents; MR, group treated with long-acting erythropoiesis-stimulating agents.

**Figure 3 jcm-12-00625-f003:**
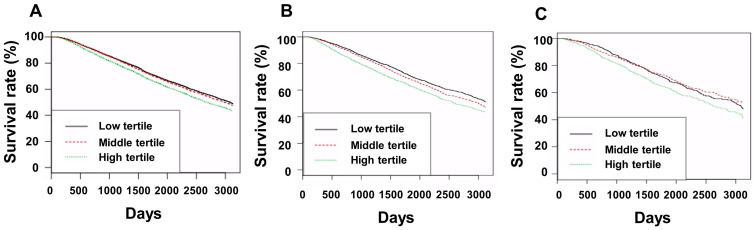
Kaplan–Meier curves of patient survival according to tertile of ESA doses in EP (**A**), DP (**B**), and MR (**C**) groups. For EP group, *p*-values were 0.045 for low vs. middle tertile, <0.001 for low vs. high tertile, and <0.001 for middle vs. high tertile. For DP group, *p*-values were 0.003 for low vs. middle tertile, <0.001 for low vs. high tertile, and <0.001 for middle vs. high tertile. For MR group, *p*-values were 0.474 for low vs. middle tertile, 0.006 for low vs. high tertile, and 0.002 for middle vs. high tertile. Abbreviations: EP, group treated with short-acting erythropoiesis-stimulating agents; DP, group treated with intermediate-acting erythropoiesis-stimulating agents; MR, group treated with long-acting erythropoiesis-stimulating agents.

**Table 1 jcm-12-00625-t001:** Patient clinical characteristics.

	EP (*n* = 38,043)	DP (*n* = 10,054)	MR (*n* = 2253)	*p*
Age (years)	60.8 ± 13.0	61.4 ± 13.1 *	62.1 ± 12.6 *^,†^	<0.001
Sex (male, %)	22,544 (59.3%)	5683 (56.5%)	1325 (58.8%)	<0.001
Hemodialysis vintage (days)	1468 ± 1618	1383 ± 1530 *	1380 ± 1548 *	<0.001
CCI score	7.7 ± 2.9	7.7 ± 2.8	8.1 ± 2.9 *^,†^	<0.001
Body mass index (kg/m^2^)	22.4 ± 3.5	22.5 ± 3.5 *	22.8 ± 3.4 *^,†^	<0.001
Underlying etiology of ESRD				<0.001
Diabetes mellitus	17,031 (44.8%)	4681 (46.6%)	1068 (47.4%)	
Hypertension	10,069 (26.5%)	2474 (24.6%)	577 (25.6%)	
Glomerulonephritis	3846 (10.1%)	1058 (10.5%)	220 (9.8%)	
Other	2951 (7.8%)	976 (9.7%)	175 (7.8%)	
Unknown	4146 (10.9%)	865 (8.6%)	213 (9.5%)	
Follow-up duration (days)	1814 ± 847	1763 ± 838 *	1826 ± 830 ^†^	<0.001
Type of vascular access				<0.001
Arteriovenous fistula	31,631 (83.1%)	8228 (81.8%)	1867 (82.9%)	
Arteriovenous graft	5502 (14.5%)	1514 (15.1%)	345 (15.3%)	
Catheter	910 (2.4%)	312 (3.1%)	41 (1.8%)	
Kt/V_urea_	1.52 ± 0.27	1.56 ± 0.29 *	1.56 ± 0.27 *	<0.001
Ultrafiltration volume (L/session)	2.26 ± 0.95	2.19 ± 0.97 *	2.16 ± 0.99 *	<0.001
Hemoglobin (g/dL)	10.6 ± 0.7	10.7 ± 0.7 *	10.7 ± 0.8 *	<0.001
Serum albumin (g/dL)	4.00 ± 0.34	3.92 ± 0.35 *	3.93 ± 0.33 *	<0.001
Serum phosphorus (mg/dL)	5.0 ± 1.4	4.8 ± 1.3 *	4.9 ± 1.4 ^†^	<0.001
Serum calcium (mg/dL)	8.9 ± 0.8	8.8 ± 0.7 *	8.8 ± 0.8 *	<0.001
Serum creatinine (mg/dL)	9.4 ± 2.7	9.3 ± 2.7 *	9.0 ± 2.6 *^,†^	<0.001
Systolic blood pressure (mmHg)	142 ± 15	140 ± 16 *	143 ± 16 ^†^	<0.001
Diastolic blood pressure (mmHg)	78 ± 9	76 ± 10 *	76 ± 10 *	<0.001

Data are expressed as mean ± standard deviation for continuous variables and as numbers (percentages) for categorical variables. *p*-values were tested using one-way analysis of variance, followed by Tukey post hoc test, and Pearson’s χ^2^ test was used for categorical variables. Abbreviations: EP, group treated with short-acting erythropoiesis-stimulating agents; DP, group treated with intermediate-acting erythropoiesis-stimulating agents; MR, group treated with long-acting erythropoiesis-stimulating agents; CCI, Charlson comorbidity index; ESRD, end-stage renal disease. * *p* < 0.05 vs. EP, ^†^ *p* < 0.05 vs. DP.

**Table 2 jcm-12-00625-t002:** Cox regression analyses of survival of HD patients.

	Univariate		Multivariate	
	HR (95% CI)	*p*	HR (95% CI)	*p*
Ref: EP group				
DP group	1.01 (0.98–1.05)	0.527	1.01 (0.97–1.05)	0.740
MR group	0.97 (0.91–1.04)	0.433	0.89 (0.83–0.97)	0.006
Ref: DP group				
MR group	0.96 (0.89–1.04)	0.302	0.89 (0.82–0.97)	0.006
Age (per 1 year increase)	1.07 (1.06–1.07)	<0.001	1.06 (1.06–1.07)	<0.001
Sex (ref: male)	0.85 (0.83–0.88)	<0.001	0.75 (0.72–0.78)	<0.001
CCI score (per 1 score increase)	1.14 (1.13–1.14)	<0.001	1.06 (1.06–1.07)	<0.001
Body mass index (per 1 kg/m^2^ increase)	0.97 (0.97–0.98)	<0.001	0.98 (0.97–0.98)	<0.001
Underlying etiology of ESRD (ref: DM)				
Glomerulonephritis	0.34 (0.32–0.37)	<0.001	0.53 (0.48–0.57)	<0.001
Hypertension	0.66 (0.64–0.69)	<0.001	0.67 (0.64–0.70)	<0.001
Other	0.53 (0.50–0.56)	<0.001	0.69 (0.64–0.74)	<0.001
Unknown	0.60 (0.57–0.64)	<0.001	0.70 (0.65–0.74)	<0.001
Hemoglobin (per 1 g/dL increase)	0.85 (0.83–0.87)	<0.001	0.94 (0.91–0.96)	<0.001
Serum creatinine (per 1 mg/dL increase)	0.86 (0.86–0.87)	<0.001	0.94 (0.94–0.95)	<0.001
SBP (per 1 mmHg increase)	1.01 (1.01–1.01)	<0.001	1.00 (1.00–1.01)	<0.001
DBP (per 1 mmHg increase)	0.98 (0.98–0.98)	<0.001	1.00 (1.00–1.01)	0.011
Serum calcium (per 1 mg/dL increase)	0.94 (0.92–0.95)	<0.001	1.07 (1.05–1.09)	<0.001
Serum phosphorus (per 1 mg/dL increase)	0.84 (0.83–0.85)	<0.001	1.04 (1.03–1.06)	<0.001
Kt/V_urea_ (per 1 unit increase)	0.91 (0.86–0.97)	0.002	0.74 (0.68–0.80)	<0.001
Serum albumin (per 1 g/dL increase)	0.36 (0.35–0.38)	<0.001	0.63 (0.60–0.67)	<0.001
HD vintage (per 1 day increase)	1.00 (1.00–1.00)	0.004	1.00 (1.00–1.00)	<0.001
Ultrafiltration volume (per 1 L increase)	0.92 (0.90–0.93)	<0.001	1.06 (1.04–1.08)	<0.001
ERI (per 1 unit increase)	1.02 (1.02–1.02)	<0.001	1.03 (1.02–1.04)	<0.001
ESA dose (per 1 unit/week increase)	1.00 (1.00–1.00)	<0.001	1.00 (1.00–1.00)	<0.001

Multivariate analysis was adjusted for group, age, sex, CCI score, body mass index, underlying etiology of ESRD, hemoglobin, serum creatinine, SBP, DBP, serum calcium, serum phosphorus, Kt/V_urea_, serum albumin, HD vintage, ultrafiltration volume, ERI, and ESA dose, and it was performed using enter mode. Abbreviations: HD, hemodialysis; HR, hazard ratio; CI, confidence interval; EP, group treated with short-acting erythropoiesis-stimulating agents; DP, group treated with intermediate-acting erythropoiesis-stimulating agents; MR, group treated with long-acting erythropoiesis-stimulating agents; CCI, Charlson comorbidity index; ESRD, end-stage renal disease; DM, diabetes mellitus; SBP, systolic blood pressure; DBP, diastolic blood pressure; ERI, erythropoietin resistance index; ESA, erythropoiesis-stimulating agent.

**Table 3 jcm-12-00625-t003:** Limitations of the study.

	Limitations
Design	Retrospective observational design
Data set	No data on iron supplementation or iron status
	No data on micronutrients
	No data on the route of injection of ESAs
	No data on patient inflammatory status during the treatment period
	No data on safety issues related to ESA treatment
	Definitions of types or doses of ESA using claims data
	Definitions of comorbidities using claims data
	No consideration of the factors related to the centers or physicians
	Use of two or more ESAs in a small proportion of patients
Statistical analysis	Imbalance in sample size and baseline characteristics among the three groups

Abbreviations: ESA, erythropoiesis-stimulating agent.

## Data Availability

The raw data were generated at the Health Insurance Review and Assessment Service. The database can be requested from the Health Insurance Review and Assessment Service by sending a study proposal that includes the purpose of the study, study design, and duration of analysis through an e-mail (turtle52@hira.or.kr) or on the website (https://www.hira.or.kr). The authors cannot distribute the data without permission.

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
