# Peer review of "Comparison of Patient Survival According to Erythropoiesis-Stimulating Agent Type of Treatment in Maintenance Hemodialysis Patients"

_jcm, 2023, doi:10.3390/jcm12020625_

Round 1

Reviewer 1 Report

Comparisons of patient survival according to erythropoiesis-stimulating agent types in maintenance hemodialysis patients

This interesting study evaluated the difference in survival among patients treated with three distinct types of ESA. The study found that MR group had a lower ERI and a better survival rate. The study deals important problems and is of wide interest to readers. However, several concerns need to be addressed.

1.      Nutritional status is one of the most important factors that affect both survival and ERI. However, BMI (using dry weight) was not included in the analysis. Authors need to include BMI as one of the covariates in the multivariate analysis.

2.      Iron status was not included in the analysis, which was appropriately noted in the limitation. Did any of the participants have active bleeding, because this is the important cause of high ERI that can potentially influence survival. Please state this point in the Methods section.

3.      In addition to iron, the study misses very important micro-nutrients that influences both resistance to ESA and survival, which is selenium and zinc. Previous studies have shown that reduced levels of these nutrients are important factors that influence the outcomes in kidney disease patients, e.g., PMID 20557491, PMID 32278520, and PMID 35812286. Authors should cite these studies and add a discussion for comprehensiveness.

4.      The reviewer is curious to know why the ERI was different among three groups. It would be informative for the readers to clarify the explanatory factors by performing additional analysis.

Author Response

This interesting study evaluated the difference in survival among patients treated with three distinct types of ESA. The study found that MR group had a lower ERI and a better survival rate. The study deals important problems and is of wide interest to readers. However, several concerns need to be addressed.

  1. Nutritional status is one of the most important factors that affect both survival and ERI. However, BMI (using dry weight) was not included in the analysis. Authors need to include BMI as one of the covariates in the multivariate analysis.

Answer: We would like to thank the reviewer for their comments. Data representing body mass index and the variable for covariates of multivariate Cox regression analyses have been included in the manuscript. Table 1 has been revised to include the baseline characteristics, and Tables 2 and S1–S7 have been revised to include body mass index in the multivariate Cox regression analyses. Similar results were obtained with and without the addition of body mass index as a covariate.

  1. Iron status was not included in the analysis, which was appropriately noted in the limitation. Did any of the participants have active bleeding, because this is the important cause of high ERI that can potentially influence survival. Please state this point in the Methods section.

Answer: We would like to express our gratitude of your kind comment. As highlighted by the reviewer, active bleeding correlates with high ERI levels. In this study, none of the patients received blood transfusions during the 6 months of HD quality assessment. Therefore, these may be considered as none of the patients with active bleeding during the period. The Methods section has been updated accordingly.

  1. In addition to iron, the study misses very important micro-nutrients that influences both resistance to ESA and survival, which is selenium and zinc. Previous studies have shown that reduced levels of these nutrients are important factors that influence the outcomes in kidney disease patients, e.g., PMID 20557491, PMID 32278520, and PMID 35812286. Authors should cite these studies and add a discussion for comprehensiveness.

 Answer: We would like to thank you for your kind comments. The articles were revised as per the suggestion. The relevant articles have been appropriately cited, and the following information has been included in the Discussion section:

Micro-nutrients influence iron status and the responsiveness of ESA and HD patients’ survival. Micronutrient levels are lower in HD patients compared to the general population. This decrease correlates with decreasing gastrointestinal absorption and losses during HD [1]. Micro-nutrients, such as zinc, selenium, or magnesium, are well-known cofactors for enzymatic systems, and they play a key role in maintaining antioxidant activity. Although the pathophysiology of micronutrients’ role in HD patients is poorly understood, epidemiological studies have shown positive association between micro-nutrients and HD patient outcomes. A cross-sectional study evaluated the association between the response to ESA and selenium level in HD patients in Japan [2]. Their study showed an inverse association in response to ESA based on selenium level, and this association was independent of iron status. A retrospective observational study, including 85 dialysis patients, showed a positive association between decreased selenium levels and mortality in dialysis patients [3].

Kobayashi et al. evaluated the importance of zinc in responsiveness of ESA in HD patients [4]. HD patients were randomly divided into two groups: (1) patients without zinc replacement and (2) patients with zinc replacement. Significant improvements in the responsiveness of ESA and iron status were observed in patients with zinc replacement compared to patients without zinc replacement. A multicenter prospective study evaluated the association between clinical outcomes and dietary zinc intake in HD patients [5]. The results showed that dietary zinc intake correlated with nutritional status, body composition, and mortality in HD patients. Other studies that evaluated the association between magnesium and mortality in dialysis patients and a meta-analysis using 21 studies revealed a positive association between the level of magnesium and all-cause or cardiovascular mortality in dialysis patients [6]. Yu et al. also showed an association between magnesium levels and the responsiveness of ESA in HD patients [7]. Previous studies also suggested that improvements in insulin sensitivity, serum albumin level, endothelial function, or energy metabolism of erythrocytes may contribute to improvement in the responsiveness of ESA by magnesium [8-10]. Our study did not include data on micronutrients. Therefore, an analysis of micronutrients is beyond of the scope of our study. Even so, we identify that the responsiveness of ESA and mortality had significant differences according to ESA type. Variations in micronutrient levels may account for the observed differences in the responsiveness of ESA and mortality in HD patients. Further studies on micronutrients are required to elucidate the cause-and-effect relationships for the observed differences in responsiveness of ESA or mortality according to ESA type.

Added reference

[1] Rucker D, Thadhani R, Tonelli M. Trace element status in hemodialysis patients. Semin Dial. 2010;23(4):389-395. 

[2] Yasukawa M, Arai S, Nagura M, et al. Selenium Associates With Response to Erythropoiesis-Stimulating Agents in Hemodialysis Patients. Kidney Int Rep. 2022;7(7):1565-1574. 

[3] Anadón Ruiz A, Martín Jiménez E, Bermejo-Barrera P, Lozano R, Seijas VM. Selenium and All-cause Mortality in End-Stage Renal Disease. Retrospective Observational Cohort Study. J Ren Nutr. 2020;30(6):484-492.

[4] Kobayashi H, Abe M, Okada K, Tei R, Maruyama N, Kikuchi F, Higuchi T, Soma M. Oral zinc supplementation reduces the erythropoietin responsiveness index in patients on hemodialysis. Nutrients. 2015 May 15;7(5):3783-95. 

[5] Garagarza C, Valente A, Caetano C, Ramos I, Sebastião J, Pinto M, Oliveira T, Ferreira A, Sousa Guerreiro C. Zinc Deficient Intake in Hemodialysis Patients: A Path to a High Mortality Risk. J Ren Nutr. 2022 Jan;32(1):87-93. 

[6] Huang CY, Yang CC, Hung KC, Jiang MY, Huang YT, Hwang JC, Hsieh CC, Chuang MH, Chen JY. Association between hypomagnesemia and mortality among dialysis patients: a systematic review and meta-analysis. PeerJ. 2022 Oct 11;10:e14203.

[7] Yu L, Song J, Lu X, Zu Y, Li H, Wang S. Association between Serum Magnesium and Erythropoietin Responsiveness in Hemodialysis Patients: A Cross-Sectional Study. Kidney Blood Press Res. 2019;44(3):354-361. 

[8] Lee S, Ryu JH, Kim SJ, Ryu DR, Kang DH, Choi KB. The Relationship between Magnesium and Endothelial Function in End-Stage Renal Disease Patients on Hemodialysis. Yonsei Med J. 2016 Nov;57(6):1446-53. 

[9] Moctezuma-Velázquez C, Gómez-Sámano MÁ, Cajas-Sánchez MB, Reyes-Molina DL, Galindo-Guzmán M, Meza-Arana CE, Cuevas-Ramos D, Gómez-Pérez FJ, Gulias-Herrero A. High Dietary Magnesium Intake is Significantly and Independently Associated with Higher Insulin Sensitivity in a Mexican-Mestizo Population: A Brief Cross-Sectional Report. Rev Invest Clin. 2017 Jan-Feb;69(1):40-46.

[10] Toprak O, Kurt H, Sarı Y, Şarkış C, Us H, Kırık A. Magnesium Replacement Improves the Metabolic Profile in Obese and Pre-Diabetic Patients with Mild-to-Moderate Chronic Kidney Disease: A 3-Month, Randomised, Double-Blind, Placebo-Controlled Study. Kidney Blood Press Res. 2017;42(1):33-42.

  1. The reviewer is curious to know why the ERI was different among three groups. It would be informative for the readers to clarify the explanatory factors by performing additional analysis.

Answer: Thank you for your comment. Linear regression analyses were performed to identify the factors associated with ERI (Table S8). Multivariate analysis showed that age, sex, CCI score, the underlying etiology of end-stage renal disease, systolic blood pressure, Kt/Vurea, HD vintage, and ESA dose were positively associated with ERI. Body mass index; hemoglobin, serum creatinine, calcium, serum phosphorus, and serum albumin levels; and ultrafiltration volume were inversely associated with ERI. Factors associated with ERI are essential issues in the management of these patients. Our results revealed that ERI is positively correlated with age, female sex, HD vintage, and comorbidities. In addition, we observed that ERI is inversely associated with direct or indirect nutritional indicators such as serum albumin, serum phosphorus, and serum creatinine levels, body mass index, and ultrafiltration volume. A recent guideline stated that factors such as iron deficiency, vitamin deficiency, inflammation, bleeding, hyperparathyroidism, or malnutrition correlate with the hypo-responsiveness of ESA (high ERI) [1]. González-Ortiz et al. showed a positive association between protein-energy wasting and ERI in HD patients [2]. Other studies showed that geriatric nutritional risk index, body composition, and sex were associated with ERI in HD patients [3-5]. Comorbidities have variable effects on the responsiveness of ESA, which are mediated through the inflammatory response and leads to increases in ERI [5]. Our results indicated a positive association between direct or indirect nutritional indicators and ERI. An inverse association between comorbidities and ERI is congruent with findings from previous studies. Although factors associated with ERI were beyond the scope of our study, our results may aid in identifying an independent factor for ERI. This information has been included in the Results and Discussion sections of the manuscript.

Added references

[1] Kidney Disease: Improving Global Outcomes (KDIGO) Anemia Work Group. KDIGO Clinical Practice Guideline for Anemia in Chronic Kidney Disease. Kidney Int Suppl. 2012;2:279-335.

[2] González-Ortiz A, Correa-Rotter R, Vázquez-Rangel A, Vega-Vega O, Espinosa-Cuevas Á. Relationship between protein-energy wasting in adults with chronic hemodialysis and the response to treatment with erythropoietin. BMC Nephrol. 2019 Aug 14;20(1):316. 

[3] Yajima T, Yajima K, Takahashi H. Association of the erythropoiesis-stimulating agent resistance index and the geriatric nutritional risk index with cardiovascular mortality in maintenance hemodialysis patients. PLoS One. 2021 Jan 15;16(1):e0245625. 

[4] Lee HY, Suh SW, Hwang JH, Shin J. Responsiveness to an erythropoiesis-stimulating agent is correlated with body composition in patients undergoing chronic hemodialysis. Front Nutr. 2022 Dec 2;9:1044895. 

[5] Mallick S, Rafiroiu A, Kanthety R, Iqbal S, Malik R, Rahman M. Factors predicting erythropoietin resistance among maintenance hemodialysis patients. Blood Purif. 2012;33(4):238-44. 

Reviewer 2 Report

.

Author Response

The “Comparisons of patient survival according to erythropoiesis stimulating agent types in maintenance hemodialysis patients” retrospective study aimed to evaluate the difference in patient survival according to the type of ESA in the Korean HD patients. The conclusions of the authors were that long acting group had comparable or better patient survival than the EP or DP groups in multivariate analysis, although it was difficult to determine whether better patient survival in the MR group originated from the type of ESA, ESA dose, ERI, or other hidden factors.

This is a large retrospective observational study with the limitations of this study design as correctly underlined by the authors themselves. I suggest to use a table for better underline these important limitation being a crucial aspect of this paper.

Answer: Thank you for your kind comment. We have added Table S9 detailing the limitations of the study, as follows:

Table S9. Limitations of the Study.

Limitations

Design

Retrospective observational design

Data set

No data on iron supplementation or iron status

No data on micro-nutrients

No data on the route of injection of ESA

Definition of types or doses of ESA using claims data

Definition of comorbidities using claims data

No consideration of the factors related to the centers or physicians

Use of two or more ESAs in a small proportion of patients

Statistical analysis

Imbalance in sample size and baseline characteristics among the three groups

Abbreviations: ESA, erythropoiesis-stimulating agent.

The study of Minutolo is a retrospective observational study in non-dialysis population: please modify the test accordingly.

Answer: Thank you for your kind comment. We have modified the test accordingly and added the required details in the manuscript per your comment.

Round 2

Reviewer 1 Report

I have no more comments.

Author Response

Thank you for reviewer comments. We hope that all comments have been adequately addressed, and the manuscript is now worthy of publication.